# Proton-Pump Inhibitors and Hypomagnesaemia in Kidney Transplant Recipients

**DOI:** 10.3390/jcm8122162

**Published:** 2019-12-06

**Authors:** Rianne M. Douwes, António W. Gomes-Neto, Joëlle C. Schutten, Else van den Berg, Martin H. de Borst, Stefan P. Berger, Daan J. Touw, Eelko Hak, Hans Blokzijl, Gerjan Navis, Stephan J. L. Bakker

**Affiliations:** 1Department of Internal Medicine, Division of Nephrology, University Medical Center Groningen, University of Groningen, 9700 RB Groningen, The Netherlands; 2Department of Clinical Pharmacy and Pharmacology, University Medical Center Groningen, University of Groningen, 9700 RB Groningen, The Netherlands; 3Unit PharmacoTherapy, -Epidemiology and -Economics, Groningen Research Institute of Pharmacy, University of Groningen, 9713 AV Groningen, The Netherlands; 4Department of Gastroenterology and Hepatology, University Medical Center Groningen, University of Groningen, 9700 RB Groningen, The Netherlands

**Keywords:** proton-pump inhibitors, magnesium, hypomagnesaemia, kidney transplantation

## Abstract

Proton-pump inhibitors (PPIs) are commonly used after kidney transplantation and there is rarely an incentive to discontinue treatment. In the general population, PPI use has been associated with hypomagnesaemia. We aimed to investigate whether PPI use is associated with plasma magnesium, 24-h urinary magnesium excretion and hypomagnesaemia, in kidney transplant recipients (KTR). Plasma magnesium and 24-h urinary magnesium excretion were measured in 686 stable outpatient KTR with a functioning allograft for ≥1 year from the TransplantLines Food and Nutrition Biobank and Cohort-Study (NCT02811835). PPIs were used by 389 KTR (56.6%). In multivariable linear regression analyses, PPI use was associated with lower plasma magnesium (β: −0.02, *P* = 0.02) and lower 24-h urinary magnesium excretion (β: −0.82, *P* < 0.001). Moreover, PPI users had a higher risk of hypomagnesaemia (plasma magnesium <0.70 mmol/L), compared with non-users (Odds Ratio (OR): 2.12; 95% confidence interval (CI) 1.43–3.15, *P* < 0.001). This risk tended to be highest among KTR taking high PPI dosages (>20 mg omeprazole Eq/day) and was independent of adjustment for potential confounders (OR: 2.46; 95% CI 1.32–4.57, *P* < 0.005). No interaction was observed between PPI use and the use of loop diuretics, thiazide diuretics, tacrolimus, or diabetes (*P*_interaction_ > 0.05). These results demonstrate that PPI use is independently associated with lower magnesium status and hypomagnesaemia in KTR. The concomitant decrease in urinary magnesium excretion indicates that this likely is the consequence of reduced intestinal magnesium absorption. Based on these results, it might be of benefit to monitor magnesium status periodically in KTR on chronic PPI therapy.

## 1. Introduction

Proton-pump inhibitors (PPIs) are frequently used after kidney transplantation for their gastro- protective properties in the setting of immunosuppressive therapy, which usually includes glucocorticoids. Since their first introduction in the late 1980s, numerous case reports and observational studies have been published that associate PPI use with unfavorable clinical outcomes, including an increased risk of hypomagnesaemia [1,2,3,4,5,6,7,8]. Recently, this observation has been strengthened by a large population based cohort study which demonstrated a two times higher risk of hypomagnesaemia among subjects from the general populations on chronic PPI therapy compared to non-users [9]. 

Magnesium homeostasis depends mainly on the balance between intestinal Mg^2+^ uptake, storage and resorption from bones and urinary excretion of Mg^2+^ via the kidneys [10]. It is postulated that PPIs induce hypomagnesaemia through inhibition of pH-dependent active magnesium absorption via transient receptor potential melastatin (TRPM) 6 and 7 channels in the intestine [11,12]. Moreover, increased renal magnesium retention has been observed in magnesium depleted subjects using chronic PPI therapy, indicating a defect in intestinal magnesium absorption or increased losses into the gastrointestinal tract, rather than renal magnesium wasting [1,7,13]. 

Hypomagnesaemia is very common after kidney transplantation and it is generally thought to be a side effect of immunosuppressive therapy, especially of calcineurin inhibitors (CNI) which are known to induce renal magnesium wasting [14]. It has been shown that hypomagnesaemia is not only present in the immediate post transplantation period, but persists in about 20% of kidney transplant recipients (KTR) for many years after transplantation [15,16]. Importantly, hypomagnesaemia has been associated with onset of post-transplant diabetes mellitus (PTDM) in KTR [17,18] and has also been associated with increased risk of cardiovascular morbidity [19,20] and mortality [21] in the general population. Whether use of PPIs contributes to hypomagnesaemia in KTR has not been well established. To our knowledge only one cohort study investigated the association between PPI use and hypomagnesaemia in 512 KTR, with negative results [22]. Reasons for absence of an association were unclear, but may have included a low prevalence of PPI use of 20%, which could have led to low statistical power of the study. Thus, whether PPI use negatively affects magnesium status after transplantation remains to be determined. We aimed to investigate whether PPI use is associated with magnesium status and hypomagnesaemia in a large single center cohort of stable outpatient KTR, in which plasma magnesium measurements were not part of routine clinical care but were assessed from samples that had been stored in a biobank. 

## 2. Methods

### 2.1. Study Design and Population 

This is a cross-sectional analysis using data from a previously described prospective cohort study registered at clinicaltrials.gov as ‘TransplantLines Food and Nutrition Biobank and Cohort-study’, NCT02811835 [23]. In summary, all adult KTR with a functioning graft beyond the first year after transplantation and without known or apparent systemic illnesses (i.e., malignancies other than cured skin cancer, opportunistic infections, overt congestive heart failure) who visited the outpatient clinic of the University Medical Center Groningen (UMCG) between November 2008 and March 2011, were asked to participate. A total of 707 out of the initially 817 invited KTR signed informed consent. We excluded KTR with missing biomaterial (*n* = 8), missing data on PPI dosage (*n* = 1), with on-demand PPI use (*n* = 3) or using magnesium supplements (*n* = 6) from statistical analyses, leaving 689 cases eligible for analysis. Study measurements were performed during a single study visit at the outpatient clinic. All study procedures were conducted in adherence with the Declaration of Helsinki and Declaration of Istanbul. The institutional review board of the UMCG approved the study protocol (METC 2008/186, approved on 17 September 2008). 

### 2.2. Exposure Definition

PPI type and daily dosage were obtained from electronic patient records and are demonstrated in Appendix A. KTR using any PPI on a daily basis during a period of at least 3 months prior to the study visit were defined as chronic PPI users as described previously [24]. To investigate a potential dose–response relationship, KTR were divided into three groups based on daily PPI dose defined in omeprazole equivalents: no PPI, low PPI dose (≤20 mg omeprazole equivalents/day (Eq/day)) and high PPI dose (>20 mg omeprazole Eq/day) [24,25].

### 2.3. Assessement of Plasma and Urinary Magnesium

Plasma magnesium was measured in samples containing lithium heparin, using a xylidyl blue method (Roche Modular analyzer, Roche Diagnostics, Mannheim, Germany). Urinary magnesium excretion was assessed in 24 h-urine samples and measured on a MEGA clinical chemistry analyzer (Merck, Darmstadt, Germany). Hypomagnesaemia was defined as plasma magnesium <0.70 mmol/L.

### 2.4. Assessment of Dietary Magnesium Intake

Dietary magnesium intake was calculated using a validated semi quantitative food frequency questionnaire (FFQ) developed and updated at the Wageningen University, which was filled out at home [26,27]. Dietary data were converted into daily nutrient intake using the Dutch Food Composition Table of 2006 [28].

### 2.5. Assessment of Covariates

Medical history was obtained from electronic patient records as described previously [23]. History of cardiovascular disease was classified according to the International Classification of Diseases, 10th revision (ICD-10) code Z86.7. Body mass index (BMI) was calculated as weight in kilograms divided by height in meters squared. Blood pressure was measured as described in detail previously [29]. Information on alcohol use and smoking behavior was obtained using a questionnaire. Medication use, including the use of PPIs, H2-receptor antagonists, diuretics, prednisolone, mycophenolate mofetil (MMF), tacrolimus, cyclosporine, and sirolimus was recorded at baseline. Routine immunosuppressive therapy consisted of: A combination of azathioprine and prednisolone from 1968 to 1989; a combination of cyclosporine and prednisolone from 1989 to 1996. In 1997 mycophenolate motefil was added to the standard immunosuppressive regimen and cyclosporine was slowly withdrawn after the first year in KTR without complications. In 2012 cyclosporine was replaced by tacrolimus, and KTR received triple-immunosuppressive therapy with prednisolone, tacrolimus and mycophenolate mofetil. PPIs were routinely prescribed after kidney transplantation for their gastro-protective properties with concurrent use of prednisolone. Blood samples were collected after an 8–12 h fasting period. Serum creatinine was measured using an enzymatic, isotope dilution mass spectrometry-traceable assay (P-Modular automated analyzer, Roche Diagnostics, Mannheim, Germany). Estimated glomerular filtration rate (eGFR) was calculated using the serum creatinine based Chronic Kidney Disease Epidemiology Collaboration (CKD-EPI) equation. Serum potassium, calcium, parathyroid hormone (PTH), glucose and hemoglobin A1c (HbA1c), were determined using standard laboratory methods. Proteinuria was defined as urinary protein excretion ≥0.5 g/24 h. 

### 2.6. Statistical Analyses

Statistical analyses were performed using SPSS, version 23.0 (IBM corp., Armonk, NY, USA). Data are presented as mean ± SD for normally distributed data, median with interquartile range (IQR) for skewed data and number with percentage for nominal data. Differences between PPI users versus PPI non-users were tested using independent sample *T*-tests, Mann–Whitney U-tests and Chi-square tests or Fishers exact tests when appropriate. 

To study the effect of PPI use on plasma magnesium linear regression analyses were performed with adjustment for potential confounders of magnesium status including: age, sex, BMI, eGFR, proteinuria, time since transplantation, alcohol use, diabetes, history of cardiovascular disease, use of loop diuretics, thiazide diuretics, tacrolimus, cyclosporine, MMF, and dietary magnesium intake. To investigate the association between PPI use and hypomagnesaemia we performed logistic regression analyses with adjustment for the same potential confounders used in multivariable linear regression analyses. Effect modification by loop diuretics, thiazide diuretics, tacrolimus and diabetes was tested by inclusion of interaction terms. To investigate a potential dose–response relationship we performed additional analyses in which KTR were divided into three groups based on daily PPI dose defined in omeprazole equivalents: No PPI, low PPI dose (≤20 mg omeprazole Eq/day) and high PPI dose (>20 mg omeprazole Eq/day) [24,25]. Tests of linear trend were conducted by assigning the median of daily PPI dose equivalents in subgroups treated as a continuous variable. We performed sensitivity analyses in which H2-receptor antagonist (H2RA) users (*n* = 18) were excluded to assess the robustness of the association between PPI use and hypomagnesaemia. Lastly, we investigated which KTR are at increased risk of developing hypomagnesaemia. A two-sided *P*-value < 0.05 was considered statistically significant in all analyses. 

## 3. Results

### 3.1. Baseline Characteristics 

Baseline characteristics are shown in Table 1. PPIs were used by a small majority of 389 (56.5%) KTR and omeprazole was the most often prescribed PPI (*n* = 340). Other PPIs used were esomeprazole (*n* = 30), pantoprazole (*n* = 16), and rabeprazole (*n* = 3). KTR who used PPIs were older than KTR who did not use PPIs, had a higher BMI and had shorter time between transplantation and baseline measurements. Diabetes was significantly more prevalent in PPI users compared with non-users (28.3% vs. 18.3%, *P* < 0.002). Plasma magnesium and 24-h urinary magnesium excretion were significantly lower in PPI users and 102 (26.2%) PPI users had hypomagnesaemia compared with 43 (14.3%) non-users (*P* < 0.001). Dietary magnesium intake was not significantly different between PPI users and non-users. Loop diuretics, cyclosporine and MMF, were more often used by PPI users compared with non-users. Triple immunosuppressive therapy consisting of MMF, cyclosporine and prednisolone, was more common in PPI users compared with non-users. Duo therapy consisting of MMF-prednisolone, MMF-cyclosporine, and cyclosporine-prednisolone was more common in PPI users compared with non-users.

### 3.2. Association of PPI Use with Plasma Magnesium and 24-h urinary Magnesium Excretion

PPI use was significantly associated with lower plasma magnesium (β = −0.03; 95% CI −0.04; −0.01 mmol/L, *P* = 0.001) and lower urinary magnesium excretion (β = −0.86; 95% CI −1.10; −0.06 mmol/24 h, *P* < 0.001) as compared to non-users, Table 2. After adjustment for potential confounders, PPI use remained significantly associated with lower plasma magnesium levels (β = −0.02, 95% CI −0.04; −0.003, *P* = 0.02) and 24-h urinary magnesium excretion (β = −0.82, 95% CI −1.07; −0.57, *P* < 0.001).

### 3.3. Association of PPI Use with Hypomagnesaemia

In crude logistic regression analysis, PPI use was associated with a more than two times higher risk of hypomagnesaemia compared with no use (OR: 2.12; 95% CI 1.43–3.15, *P* < 0.001), as shown in Table 3. The association remained independent of adjustment for potential confounders including age, sex, eGFR, proteinuria, time since transplantation, alcohol use, diabetes, history of cardiovascular disease, medication use (loop diuretics, thiazide diuretics, tacrolimus, cyclosporine and MMF) and dietary magnesium intake (OR: 2.00; 95% CI 1.21–3.31, *P* = 0.007). No significant interaction was observed between PPI use and the use of loop diuretics, thiazide diuretics, tacrolimus, or diabetes for the association with hypomagnesaemia (*P*_interaction_ = 0.2, *P*_interaction_ = 0.7, *P*_interaction_ = 0.7, *P*_interaction_ = 0.9, respectively).

### 3.4. Dose–Response Analyses

Based on daily dose equivalents of omeprazole, 251 KTR received a low PPI dose (≤20 mg omeprazole Eq/day) and 138 KTR received a high PPI dose (>20 mg omeprazole Eq/day). As shown in Table 4 and Figure 1, risk of hypomagnesaemia tended to be highest among KTR taking a high PPI dose (OR: 2.53; 95% CI 1.55–4.11, *P* < 0.001). The association remained materially unchanged after multivariable adjustment (OR: 2.46; 95% CI 1.32–4.57, *P* < 0.005), Table 4. Moreover, a significant trend between PPI dose and risk of hypomagnesaemia was observed (*P*_trend_ = 0.004).

### 3.5. Sensitivity Analyses for Risk of Hypomagnesaemia

To account for the use of other important gastric acid reducing medication, we performed sensitivity analyses in which H2RA users (*N* = 18) were excluded form statistical analyses (Appendix A). The association between PPI use and hypomagnesaemia remained materially unchanged when H2RA users were excluded (OR: 2.17, 95% CI 1.29–3.67, *P* = 0.004). We also performed analyses to investigate which KTR are at increased risk of developing hypomagnesaemia. These analyses are presented in Appendix A. We found that patients with a history of cardiovascular disease, patients at shorter time after transplantation, not consuming alcohol, PPI users, thiazide diuretic users and patients using tacrolimus based immunosuppressive regimens were at increased risk of developing hypomagnesaemia. Moreover, KTR with hypomagnesaemia had higher fasting glucose levels, HbA1c and lower serum calcium levels compared with KTR without hypomagnesaemia. 

## 4. Discussion

The present study is to our knowledge the largest cohort study to date exploring the association between PPI use and hypomagnesaemia in a cohort of KTR. Our results demonstrate a higher risk of hypomagnesaemia among KTR using PPIs, with subsequently lower plasma magnesium levels in combination with lower renal magnesium excretion. The association between PPI use and risk of hypomagnesaemia remained significant after adjustment for important potential confounders and tended to be highest among KTR taking high PPI dosages.

Our results confirm previous case-series and cohort studies investigating the association between PPI use and increased risk of hypomagnesaemia [1,2,7,9]. In a large cohort study (*N* = 9818) among subjects from the general population, it was shown that PPI users had significantly lower serum magnesium levels and had a two times higher risk of hypomagnesaemia compared with non-users [9]. Our results are in line with observations from this large cohort study and show a similar increased risk of hypomagnesaemia (OR 2.12).

So far, only one other study by van Ende et al. investigating the association between PPI use and magnesium status in KTR has been published [22]. Contrary to our findings, van Ende et al. found no association between PPI use and serum magnesium levels. Reasons for the lower proportion of PPI users in the study by van Ende et al. are unclear, though underreporting may have played a role, given that it was not specified how data regarding PPI use was obtained. It was also unclear whether the data of van Ende et al. were derived from routine outpatient assessment of plasma magnesium concentrations, which may have provided an incentive for stopping PPI use in KTR with low magnesium concentrations. This could have biased their results and could possibly also explain the large difference in PPI use between our study and their study, because in our center no plasma magnesium data were available at the time of the study. It was furthermore unclear whether it concerned on-demand or chronic PPI use. Furthermore, data regarding PPI dose, type and magnesium supplementation were not reported, which may have influenced the outcome.

In a recently published meta-analysis, a similar risk of hypomagnesaemia among KTR was demonstrated (pooled OR = 1.56, 95% CI 1.19–2.05) [30]. This meta-analysis by Boonpheng et al. was based on one published paper and seven abstracts presented at medical conferences. Our study adds that it investigated a dose–response relationship, and provides data on dietary magnesium intake and 24-h urinary magnesium excretion. 

In the present study, both plasma magnesium and 24-h urinary magnesium excretion were lower in PPI users, suggesting that PPI induced hypomagnesaemia is caused by impaired gastrointestinal absorption rather than renal magnesium wasting. In general, hypomagnesaemia can be the consequence of either a decreased intestinal uptake, a decrease in dietary magnesium intake or an increase in renal magnesium excretion. It is postulated that PPIs inhibit the active magnesium absorption via the TRPM 6 and 7 channels in the intestine [11,12]. In KTR other contributing factors than PPI use may add to the risk of hypomagnesaemia. For example, decreased intestinal magnesium absorption can also be the consequence of chronic post-transplant diarrhea, which is highly prevalent and often complicated by hypomagnesaemia [10,31]. Data regarding symptoms of severe diarrhea were unfortunately unavailable in this study, therefore we could not correct for this potential confounder. Likewise, hypomagnesaemia can be the result of insufficient intake of foods rich in magnesium. In our study, mean dietary magnesium intake was 329.9 ± 88.7 mg/day, which was slightly lower than the mean habitual intake of magnesium among the general Dutch population, as reported in the Dutch National Food Consumption Survey 2007–2010 [32]. A low dietary magnesium intake can also be a reflection of an overall poor diet. Nonetheless, when we adjusted for dietary magnesium intake in our logistic regression analyses, the relationship between PPI use and risk of hypomagnesaemia remained materially unchanged, indicating that the observed risk associated with PPI use was not confounded by dietary magnesium intake. 

The main strength of this study is measurement of three important pillars of magnesium status: plasma magnesium, 24-h urinary magnesium excretion and dietary magnesium intake. Because of this, we were able to confirm that PPI use does not lead to increased renal magnesium wasting but very likely impairs intestinal magnesium absorption. Furthermore, we only included KTR who were using PPIs for at least 3 months before blood sampling. It is previously noted that hypomagnesaemia occurs mainly in patients on prolonged PPI therapy suggesting that it takes time before magnesium stores are meaningfully depleted [6,7,33]. Moreover, we excluded KTR using magnesium supplements and adjusted for potential confounders, including CNI use, which did not alter the association. 

A limitation of our study is its cross-sectional design. Therefore, a causal relationship between PPI use and hypomagnesaemia remains to be determined and changes over time in magnesium status parameters were unknown. Furthermore, no information regarding compliance to PPI treatment was available, which may have led to underestimation of effect sizes. PPI users had a shorter time between transplantation and baseline measurements. However, adjustment for time since transplantation did not alter the association between PPI use and hypomagnesaemia. Lastly, the possibility of residual confounding or bias by indication remains, which may have led to overestimation of the role of PPIs since on average PPI users were less healthy than non-users. A strength of the current study is, that no routine outpatient monitoring of plasma magnesium was performed and that we measured plasma and urine magnesium in samples that had been stored in a biobank, which reduces the change of selection bias in our cohort. 

Our findings may be of clinical importance. KTR with low magnesium levels seem to develop post-transplant diabetes mellitus (PTDM) more frequently [17]. In this study we also found that KTR with hypomagnesaemia had higher fasting glucose levels and HbA1c. Next to that, a higher degree of arterial stiffness, as assessed by a carotid-femoral pulse wave velocity (PWV) measurement, has been found in KTR with low magnesium levels [34]. This same PWV measurement was found to be an independent predictor of cardiovascular events in KTR [35]. Moreover, hypomagnesaemia has been associated with cardiovascular morbidity [19,20] and mortality [21] in the general population. However, whether this association is also present in KTR is currently unknown. Another clinical significance lies in the association with lower calcium levels, which potentially points to an increased risk of developing osteoporosis. Long-term PPI use has indeed been associated with decreased bone mineral density and increased risk of fractures [36]. Because many patients use PPIs without evidence based indication [37,38,39], we believe that reevaluation of treatment indication in KTR on chronic PPI therapy might be of benefit. In situations in which PPIs are clinically needed, it would be judicious to assess and follow-up magnesium levels periodically during treatment, as recommended by the US Food and Drug administration and stated in the summary of product characteristics of all PPIs. 

## 5. Conclusions

This study demonstrates that PPI use is associated with lower magnesium status and hypomagnesaemia in KTR. Moreover, risk of hypomagnesaemia was higher among KTR taking a high PPI dosage. Healthcare professionals should be aware of this additional risk and should consider regular monitoring of magnesium levels, especially in this patient population at high risk of hypomagnesaemia. 

## Figures and Tables

**Figure 1 jcm-08-02162-f001:**
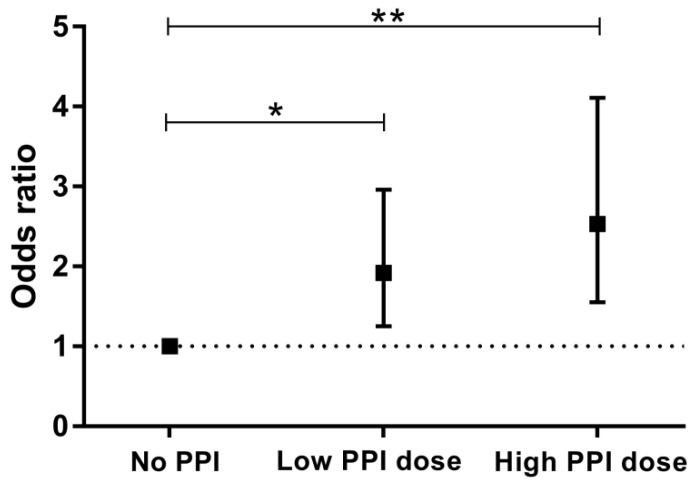
Crude association between PPI use and risk of hypomagnesaemia stratified by subgroups of PPI use. No PPI, Low PPI dose (≤20 mg omeprazole Eq/day), High PPI dose (>20 mg omeprazole Eq/day). Presented are odds ratio’s with 95% confidence intervals. * *P* = 0.004; ** *P* < 0.001; *P*_trend_ < 0.001.

**Table 1 jcm-08-02162-t001:** Baseline characteristics of 689 kidney transplant recipients.

Characteristics	Total Population	Non-PPI Users	PPI Users	*P*
Number of subjects, *n* (%)	689 (100)	300 (43.5)	389 (56.5)	n/a
Demographics				
Age, year	53 ± 13	51 ± 13	54 ± 12	0.001
Men, *n* (%)	395 (57.3)	177 (59.0)	218 (56.0)	0.4
BMI, kg/m^2^	26.6 ± 4.8	25.9 ± 4.6	27.1 ± 4.8	0.002
Diabetes Mellitus, *n* (%)	165 (23.9)	55 (18.3)	110 (28.3)	0.002
History of CV disease, *n* (%)	274 (39.8)	92 (30.7)	182 (46.8)	<0.001
Time since transplantation, year	5.5 (1.9–12.1)	9.6 (4.1–15.0)	4.2 (1.1–8.7)	<0.001
Lifestyle parameters				
Current smoker, *n* (%)	84 (13.0)	35 (12.4)	49 (13.6)	0.7
Alcohol consumer, *n* (%)	436 (70.3)	198 (72.8)	238 (68.4)	0.2
Magnesium intake, mg/day	329.9 ± 88.7	333.0 ± 89.2	327.6 ± 88.4	0.5
Renal function parameters				
eGFR, mL/min/1.73 m^2^	52.3 ± 20.2	55.1 ± 19.9	50.2 ± 20.1	0.002
Serum creatinine, µmol/L	124 (100–160)	119 (98–152)	128 (101–168)	0.03
Proteinuria (≥0.5 g/24 h), *n* (%)	157 (22.9)	71 (23.7)	86 (22.2)	0.7
Laboratory parameters				
Hypomagnesaemia, *n* (%)	145 (21.0)	43 (14.3)	102 (26.2)	<0.001
Plasma magnesium, mmol/L	0.77 ± 0.11	0.79 ± 0.09	0.76 ± 0.11	<0.001
24-h urinary magnesium excretion, mmol/24 h	3.3 (2.3–3.3)	3.8 (2.8–4.8)	3.1 (2.0–3.9)	<0.001
Serum potassium, mmol/L	3.98 ± 0.46	3.97 ± 0.47	3.99 ± 0.46	0.6
Serum calcium, mmol/L	2.40 ± 0.15	2.40 ± 0.15	2.40 ± 0.15	0.8
PTH, _p_mol/L	9.0 (6.0–14.8)	8.7 (6.0–13.6)	9.2 (5.9–16.3)	0.2
Glucose, mmol/L	5.3 (4.8–6)	5.2 (4.7–5.8)	5.3 (4.8–6.2)	0.01
HbA1c, mmol/mol	40 (37–44)	39 (36–42)	41 (38–45)	<0.001
Medication use				
Mycophenolate mofetil, *n* (%)	452 (65.6)	178 (59.3)	274 (70.4)	0.002
Tacrolimus, *n* (%)	124 (18.0)	49 (16.3)	75 (19.3)	0.3
Cyclosporine, *n* (%)	272 (39.5)	97 (32.3)	175 (45.0)	0.001
Sirolimus, *n* (%)	13 (2.0)	8 (2.8)	5 (1.4)	0.3
Prednisolone, *n* (%)	682 (99.0)	298 (99.3)	384 (98.7)	0.7
Loop diuretics, *n* (%)	160 (23.2)	41 (13.7)	119 (30.6)	<0.001
Thiazide diuretics, *n* (%)	120 (17.4)	53 (17.7)	67 (17.4)	0.9
H2-receptor antagonists, *n* (%)	18 (2.6)	17 (5.7)	1 (0.3)	<0.001
Combination therapy				
MMF + Tac + pred, *n* (%)	78 (11.3)	32 (10.7)	46 (11.8)	0.6
MMF + Cyclo + pred, *n* (%)	175 (25.4)	51 (17.0)	124 (31.9)	<0.001
MMF + Tac, *n* (%)	81 (11.8)	33 (11.0)	48 (12.3)	0.6
MMF + Pred, *n* (%)	447 (64.9)	176 (58.7)	271 (69.7)	0.003
MMF + Cyclo, *n* (%)	177 (25.7)	52 (17.3)	125 (32.1)	<0.001
Cyclo + Pred, *n* (%)	269 (39.0)	96 (32.0)	173 (44.5)	0.001
Tac + Pred, *n* (%)	120 (17.4)	48 (16.0)	72 (18.5)	0.4

Data are presented as mean ± SD, median with interquartile ranges (IQR) or number with percentages (%). Abbreviations: BMI, body mass index; eGFR, estimated glomerular filtration rate; HbA1c, hemoglobin A1c; PTH, Parathyroid hormone; MMF, mycophenolate mofetil; Tac, tacrolimus; Pred, prednisolone.

**Table 2 jcm-08-02162-t002:** Association of proton-pump inhibitor (PPI) use with plasma magnesium and 24-h urinary magnesium excretion in 689 kidney transplant recipients.

	Plasma Magnesium, mmol/L			Urinary Magnesium Excretion, mmol/24 h		
	β	95% CI	*P*	β	95% CI	*P*
Crude	−0.03	−0.04; −0.01	0.001	−0.86	−1.10; −0.06	<0.001
Multivariable model	−0.02	−0.04; −0.003	0.02	−0.82	−1.07; −0.57	<0.001

Multivariable analyses were adjusted for age, sex, BMI, eGFR, proteinuria, time since transplantation, alcohol use, diabetes, history of CV disease, loop diuretics, thiazide diuretics, tacrolimus use, cyclosporine use, MMF use and dietary magnesium intake. Abbreviations: CI, confidence interval.

**Table 3 jcm-08-02162-t003:** Logistic regression analyses investigating the association of PPI use with hypomagnesaemia in 689 kidney transplant recipients.

	Hypomagnesaemia		
*N* = 689	Odds Ratio	95% CI	*P*
Crude	2.12	1.43–3.15	<0.001
Multivariable model	2.00	1.21–3.31	0.007

Multivariable analyses were adjusted for age, sex, BMI, eGFR, proteinuria, time since transplantation, alcohol use, diabetes, history of CV disease, loop diuretics, thiazide diuretics, tacrolimus use, cyclosporine use, MMF use and dietary magnesium intake. Abbreviations: CI, confidence interval.

**Table 4 jcm-08-02162-t004:** Subgroup analyses of the association of PPI use with hypomagnesaemia in 689 kidney transplant recipients.

	Categories of PPI Use						
	No PPI		Low PPI Dose		High PPI Dose		
**Number of subjects**	300		251		138		
	Odds ratio (95% CI)	*P*	Odds ratio (95% CI)	*P*	Odds ratio (95% CI)	*P*	*P*-Trend
**Hypomagnesaemia**							
Crude	1.00 (reference)	n/a	1.92 (1.25–2.96)	0.003	2.53 (1.55–4.11)	<0.001	<0.001
Multivariable model	1.00 (reference)	n/a	1.79 (1.04–3.08)	0.04	2.46 (1.32–4.57)	0.005	0.004

Multivariable analyses were adjusted for age, sex, BMI, eGFR, proteinuria, time since transplantation, alcohol use, diabetes, history of CV disease, loop diuretics, thiazide diuretics, tacrolimus use, cyclosporine use, MMF use, dietary magnesium intake. Low PPI dose (≤20 mg omeprazole Eq/day), High PPI dose (>20 mg omeprazole Eq/day). Abbreviations: CI, confidence interval.

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
