# Peer review of "Proton-Pump Inhibitors and Hypomagnesaemia in Kidney Transplant Recipients"

_jcm, 2019, doi:10.3390/jcm8122162_

Round 1

Reviewer 1 Report

Since 2006, when the first two cases of PPIs induced hypomagnesemia  were reported, numerous case reports and series have confirmed this association and PPIs use has now also been related to hypomagnesemia in the general population. A significant association between PPIs use and hypomagnesemia has also been found in patients with chronic kidney disease at any stage. Moreover, a recent meta analysis (Boonpheng B 2019) showed that the risk of hypomagnesemia was significantly increased with PPI use in kidney transplant recipients. It is also well known that diminished intestinal magnesium absorption has been proposed as the main underlying  mechanism of PPIs related hypomagnesemia. Consequently, the under review study, lacks novelty and does not add new knowledge in the medical literature. On the contrary, potential interesting findings, such as which kidney transplant recipients have an increased risk in developing hypomagnesemia as well as the clinical significance of that side effect, are not provided. 

Reviewer 2 Report

It would be of interest to calculate differences between PPI users and Non-PPI users not only with different immunosuppressive medications but also combinations of medications (especially in combination Tacrolimus-MMF-prednisolon and ciclosporine-MMF-prednislolone).

Reviewer 3 Report

The authors present a well-conceived, well-designed, and well-executed study, clearly demonstrating an (often overlooked) association between chronic PPI use and hypomagnesemia in KTRs. Hopefully this conclusive study will raise clinical awareness of this non-trivial issue. The methodology is top-notch; the results are unassailable; and the discussion is well-reasoned. There are just a few very minor grammatical and formatting edits required. Otherwise, I have no further questions or suggestions. Well done!

Round 2

Reviewer 1 Report

The manuscript has considerably been improved.